# ONLINE ADVERSARIAL ATTACKS

**Andjela Mladenovic**[*]
Mila, Université de Montréal

**Avishek Joey Bose**[*]
Mila, McGill University

**Hugo Berard**[*†]
Mila, Université de Montréal

**William L. Hamilton**[‡]
Mila, McGill University

**Simon Lacoste-Julien**[‡]
Mila, Université de Montréal

**Pascal Vincent**[‡]
Mila, Université de Montréal
Meta AI Research

**Gauthier Gidel**[‡]
Mila, Université de Montréal

## ABSTRACT

Adversarial attacks expose important vulnerabilities of deep learning models, yet little attention has been paid to settings where data arrives as a stream. In this paper, we formalize the online adversarial attack problem, emphasizing two key elements found in real-world use-cases: attackers must operate under partial knowledge of the target model, and the decisions made by the attacker are irrevocable since they operate on a transient data stream. We first rigorously analyze a deterministic variant of the online threat model by drawing parallels to the well-studied $k$-secretary problem in theoretical computer science and propose VIRTUAL+, a simple yet practical online algorithm. Our main theoretical result shows VIRTUAL+ yields provably the best competitive ratio over all single-threshold algorithms for $k < 5$—extending the previous analysis of the $k$-secretary problem. We also introduce the *stochastic k-secretary*—effectively reducing online blackbox transfer attacks to a $k$-secretary problem under noise—and prove theoretical bounds on the performance of VIRTUAL+ adapted to this setting. Finally, we complement our theoretical results by conducting experiments on MNIST, CIFAR-10, and Imagenet classifiers, revealing the necessity of online algorithms in achieving near-optimal performance and also the rich interplay between attack strategies and online attack selection, enabling simple strategies like FGSM to outperform stronger adversaries.

## 1 INTRODUCTION

In adversarial attacks, an attacker seeks to maliciously disrupt the performance of deep learning systems by adding small but often imperceptible noise to otherwise clean data (Szegedy et al., 2014; Goodfellow et al., 2015). Critical to the study of adversarial attacks is specifying the threat model Akhtar & Mian (2018), which outlines the adversarial capabilities of an attacker and the level of information available in crafting attacks. Canonical examples include the *whitebox* threat model Madry et al. (2017), where the attacker has complete access, and the less permissive *blackbox* threat model where an attacker only has partial information, like the ability to query the target model (Chen et al., 2017; Ilyas et al., 2019; Papernot et al., 2016).

Previously studied threat models (e.g., whitebox and blackbox) implicitly assume a static setting that permits full access to instances in a target dataset at all times (Tramèr et al., 2018). However, such an assumption is unrealistic in many real-world systems. Countless real-world applications involve streaming data that arrive in an online fashion (e.g., financial markets or real-time sensor networks). Understanding the feasibility of adversarial attacks in this *online* setting is an essential question.

As a motivating example, consider the case where the adversary launches a man-in-the-middle attack depicted in Fig. 1. Here, data is streamed between two endpoints—i.e., from sensors on an autonomous car to the actual control system. An adversary, in this example, would intercept the

---

[*]Equal Contribution. Corresponding authors: {`joey.bose,andjela.mladenovic`}`@mila.quebec`
[†]Work done while an intern at Meta AI Research
[‡]Canada CIFAR AI Chair

sensor data, potentially perturb it, and then send it to the controller. Unlike classical adversarial attacks, such a scenario presents two key challenges that are representative of all online settings.

1. **Transiency:** At every time step, the attacker makes an irrevocable decision on whether to attack, and if she fails, or opts not to attack, then that datapoint is no longer available for further attacks.
2. **Online Attack Budget:** The adversary—to remain anonymous from stateful defenses —is restricted to a small selection budget and must optimally balance a passive exploration phase before selecting high-value items in the data stream (e.g. easiest to attack) to submit an attack on.

To the best of our knowledge, the only existing approaches that craft adversarial examples on streaming data (Gong et al., 2019a; Lin et al., 2017; Sun et al., 2020) require multiple passes through a data stream and thus cannot be applied in a realistic online setting where an adversary is forced into irrevocable decisions. Moreover, these approaches do not come with theoretical guarantees. Consequently, assessing the practicality of adversarial attacks—to better expose risks—in a truly online setting is still an open problem, and the focus of this paper.

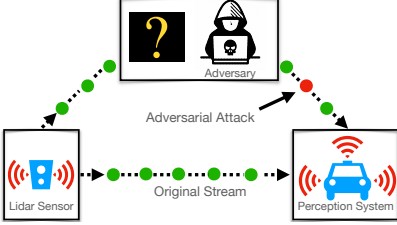

Figure 1: Man-in-the-Middle Attack.

**Main Contributions**. We formalize the online threat model to study adversarial attacks on streaming data. In our online threat model, the adversary must execute $k$ successful attacks within $n$ streamed data points, where $k \ll n$. As a starting point for our analysis, we study the deterministic online threat model in which the actual value of an input—i.e., the likelihood of a successful attack—is revealed along with the input. Our first insight elucidates that such a threat model, modulo the attack strategy, equates to the $k$-secretary problem known in the field of optimal stopping theory Dynkin (1963); Kleinberg (2005), allowing for the application of established online algorithms for picking optimal data points to attack. We then propose a novel online algorithm VIRTUAL+ that is both practical, simple to implement for any pair $(k, n)$, and requires no additional hyperparameters.

Besides, motivated by attacking blackbox target models, we also introduce a modified secretary problem dubbed the *stochastic $k$-secretary problem*, which assumes the values an attacker observes are stochastic estimates of the actual value. We prove theoretical bounds on the competitive ratio—under mild feasibility assumptions—for VIRTUAL+in this setting. Guided by our theoretical results, we conduct a suite of experiments on both toy and standard datasets and classifiers (i.e., MNIST, CIFAR-10, and Imagenet). Our empirical investigations reveal two counter-intuitive phenomena that are unique to the online blackbox transfer attack setting: 1.) In certain cases attacking robust models may in fact be easier than non-robust models based on the distribution of values observed by an online algorithm. 2.) Simple attackers like FGSM can seemingly achieve higher online attack transfer rates than stronger PGD-attackers when paired with an online algorithm, demonstrating the importance of carefully selecting which data points to attack. We summarize our key contributions:

- We formalize the online adversarial attack threat model as an online decision problem and rigorously connect it to a generalization of the k-secretary problem.
- We introduce and analyze VIRTUAL+, an extension of VIRTUAL for the $k$-secretary problem yielding a significant practical improvement (60%).
  We then provide, via novel techniques, a tractable formula for its competitive ratio, partially answering one of Albers & Ladewig (2020)'s open questions (see footnote [1]) and achieving a new state-of-the-art competitive ratio for $k < 5$.
- We propose Alg. 2 that leverages (secretary) online algorithms to perform efficient online adversarial attacks. We compare different online algorithms including VIRTUAL+ on MNIST, CIFAR-10, and Imagenet in the challenging Non-Interactive BlackBox transfer (NoBox) setting.

## 2 BACKGROUND AND PRELIMINARIES

**Classical Adversarial Attack Setup**. We are interested in constructing adversarial examples against some fixed target classifier $f_t : \mathcal{X} \to \mathcal{Y}$ which consumes input data points $x \in \mathcal{X}$ and labels them with a class label $y \in \mathcal{Y}$. The goal of an adversarial attack is then to produce an adversarial example $x' \in \mathcal{X}$, such that $f_t(x') \neq y$, and where the distance $d(x, x') \leq \gamma$. Then, equipped with a loss $\ell$ used to evaluate $f_t$, an attack is said to be optimal if (Carlini & Wagner, 2017; Madry et al., 2017),

$$x' \in \operatorname{argmax}_{x' \in \mathcal{X}} \ell(f_t(x'), y), \quad \text{s.t. } d(x, x') \leq \gamma. \tag{1}$$

Note that the formulation above makes no assumptions about access and resource restrictions imposed upon the adversary. Indeed, if the parameters of $f_t$ are readily available, we arrive at the familiar whitebox setting, and problem in Eq. 1 is solved by following the gradient $\nabla_x f_t$ that maximizes $\ell$.

**k-Secretary Problem**. The secretary problem is a well-known problem in theoretical computer science Dynkin (1963); Ferguson et al. (1989). Suppose that we are tasked with hiring a secretary from a randomly ordered set of $n$ potential candidates to select the secretary with maximum value. The secretaries are interviewed sequentially and reveal their actual value on arrival. Thus, the decision to accept or reject a secretary must be made immediately, irrevocably, and without knowledge of future candidates. While there exist many generalizations of this problem, in this work, we consider one of the most canonical generalizations known as the *k-secretary problem* Kleinberg (2005). Here, instead of choosing the best secretary, we are tasked with choosing $k$ candidates to maximize the expected sum of values. Typically, online algorithms that attempt to solve secretary problems are evaluated using the competitive ratio, which is the value of the objective achieved by an online algorithm compared to an optimal value of the objective that is achieved by an ideal "offline algorithm," i.e., an algorithm with access to the entire candidate set. Formally, an online algorithm $\mathcal{A}$ that selects a subset of items $S_\mathcal{A}$ is said to be $C$-competitive to the optimal algorithm OPT which greedily selects a subset of items $S^*$ while having full knowledge of all $n$ items, if asymptotically in $n$

$$\mathbb{E}_{\pi \sim \mathcal{S}_n}[\mathbb{V}(S_\mathcal{A})] \geq (C + o(1))\mathbb{V}(S^*),\tag{2}$$

where $\mathbb{V}$ is a set-value function that determines the sum utility of each algorithm's selection, and the expectations are over permutations sampled from the symmetric group of $n$ elements, $\mathcal{S}_n$, acting on the data. In §4, we shall further generalize the $k$-secretary problem to its stochastic variant where the online algorithm is no longer privy to the actual values but must instead choose under uncertainty.

## 3 ONLINE ADVERSARIAL ATTACKS

Motivated by our more realistic threat model, we now consider a novel adversarial attack setting where the data is no longer static but arrives in an online fashion.

### 3.1 ADVERSARIAL ATTACKS AS SECRETARY PROBLEMS

The defining feature of the online threat model—in addition to streaming data and the fact that we may not have access to the target model $f_t$—is the online attack budget constraint. Choosing when to attack under a fixed budget in the online setting can be related to a secretary problem. We formalize this online adversarial attack problem in the boxed online threat model below.

In the online threat model we are given a data stream $\mathcal{D} = \{(x_1, y_1), \ldots, (x_n, y_n)\}$ of $n$ samples ordered by their time of arrival. In order to craft an attack against the target model $f_t$, the adversary selects, using its online algorithm $\mathcal{A}$, a subset $S_\mathcal{A} \subset \mathcal{D}$ of items to maximize:

$$\mathbb{V}(S_\mathcal{A}) := \sum_{(x,y) \in S_\mathcal{A}} \ell(f_t(\text{ATT}(x)), y) \text{ s.t. } |S_A| \leq k,\tag{3}$$

where $\text{ATT}(x)$ denotes an attack on $x$ crafted by a *fixed* attack method $\text{ATT}$ that might or might not depend on $f_t$. From now on we define $x_i' = \text{ATT}(x_i)$. Intuitively, the adversary chooses $k$ instances that are the "easiest" to attack, i.e. samples with the highest value. Note that selecting an instance to attack does not guarantee a successful attack. Indeed, a successful attack vector may not exist if the perturbation budget $\gamma$ is too small. However, stating the adversarial goal as maximizing the value of $S_\mathcal{A}$ leads to the measurable objective of calculating the ratio of successful attacks in $S_\mathcal{A}$ versus $S^*$.

If the adversary knows the true value of a datapoint then the online attack problem reduces to the original $k$-secretary. On the other hand, the adversary might not have access to $f_t$, and instead, the adversary's value function may be an estimate of the true value—e.g., the loss of a surrogate classifier, and the adversary must make selection decisions in the face of uncertainty. The theory developed in this paper will tackle both the case where values $v_i := \ell(f_t(x_i'), y_i)$ for $i \in \{1, \ldots, n\} := [n]$ are known (§3.2), as well as the richer stochastic setting with only estimates of $v_i$, $i \in [n]$ (§4).

**Practicality of the Online Threat Model**. It is tempting to consider whether in practice the adversary should forego the online attack budget and instead attack every instance. However, such a strategy poses several critical problems when operating in real-world online attack scenarios. Chiefly, attacking any instance in $\mathcal{D}$ incurs a non-trivial risk that the adversary is detected by a defense mechanism.

Indeed, when faced with stateful defense strategies (e.g. Chen et al. (2020)), every additional attacked instance further increases the risk of being detected and rendering future attacks impotent. Moreover, attacking every instance may be infeasible computationally for large $n$ or impractical based on other real-world constraints. Generally speaking, as conventional adversarial attacks operate by restricting the perturbation to a fraction of the maximum possible change (e.g., $\ell_\infty$-attacks), online attacks analogously restrict the time window to a fraction of possible instances to attack. Similarly, knowledge of $n$ is also a factor that the adversary can easily control in practice. For example, in the autonomous control system example, the adversary can choose to be active for a short interval—e.g., when the autonomous car is at a particular geospatial location—and thus set the value for $n$.

> **Online Threat Model.** *The online threat model relies on the following key definitions:*
>
> - ***The target model*** $f_t$. *The adversarial goal is to attack some target model* $f_t : \mathcal{X} \rightarrow \mathcal{Y}$, *through adversarial examples that respect a chosen distance function, d, with tolerance* $\gamma$.
> - ***The data stream*** $\mathcal{D}$. *The data stream* $\mathcal{D}$ *contains the* $n$ *examples* $(x_i, y_i)$ *ordered by their time of arrival. At any timestep* $i$, *the adversary receives the corresponding item in* $\mathcal{D}$ *and must decide whether to execute an attack or forever forego the chance to attack this item.*
> - ***Online attack budget*** $k$. *The adversary is limited to a maximum of* $k$ *attempts to craft attacks within the online setting, thus imposing that each attack is on a unique item in* $\mathcal{D}$.
> - ***A value function*** $\mathcal{V}$. *Each item in the dataset is assigned a value on arrival by the value function* $\mathcal{V} : \mathcal{X} \times \mathcal{Y} \rightarrow \mathbb{R}_+$ *which represents the utility of selecting the item to craft an attack. This can be the likelihood of a successful attack under* $f_t$ *(true value) or a stochastic estimate of the incurred loss given by a surrogate model* $f_s \approx f_t$.
>
> *The online threat model corresponds to the setting where the adversary seeks to craft adversarial attacks (i) against a target model* $f_t \in \mathcal{F}$, *(ii) by observing items in* $\mathcal{D}$ *that arrive online, (iii) and choosing* $k$ *optimal items to attack by relying on (iv) an available value function* $\mathcal{V}$. *The adversary's objective is then to use its value function towards selecting items in* $\mathcal{D}$ *that maximize the sum total value of selections* $\mathbb{V}$ *(Eq. 3).*

### 3.2 VIRTUAL+ FOR ADVERSARIAL SECRETARY PROBLEMS

Let us first consider the deterministic variant of the online threat model, where the true value is known on arrival. For example consider the value function $\mathcal{V}(x_i, y_i) = \ell(f_t(x_i'), y_i) = v_i$ i.e. the loss resulting from the adversary corrupting incoming data $x_i$ into $x_i'$. Under a fixed attack strategy, the selection of high-value items from $\mathcal{D}$ is exactly the original $k$-secretary problem and thus the adversary may employ any $\mathcal{A}$ that solves the original $k$-secretary problem.

Well-known single threshold-based algorithms that solve the $k$-secretary problem include the VIR-TUAL, OPTIMISTIC Babaioff et al. (2007) and the recent SINGLE-REF algorithm Albers & Ladewig (2020). In a nutshell, these online algorithm consists of two phases—a *sampling phase* followed by a *selection phase*—and an optimal stopping point $t$ (threshold) that is used by the algorithm to transition between the phases. In the sampling phase, the algorithms passively observe all data points up to a pre-specified threshold $t$. Note that $t$ itself is algorithm-specific and can be chosen by solving a separate optimization problem. Additionally, each algorithm also maintains a sorted reference list $R$ containing the top-$k$ elements. Each algorithm then executes the selection phase through comparisons of incoming items to those in $R$ and possibly updating $R$ itself in the process (see §D).

Indeed, the simple structure of both the VIRTUAL and OPTIMISTIC algorithms—e.g., having few hyperparameters and not requiring the algorithm to involve Linear Program's for varying values of $n$ and $k$—in addition to being $(1/e)$-competitive (optimal for $k = 1$) make them suitable candidates for solving Eq. 3. However, the competitive ratio of both algorithms in the small $k$ regime—but not $k = 1$—has shown to be sub-optimal with SINGLE-REF provably yielding larger competitive ratios at the cost of an additional hyperparameter selected via combinatorial optimization when $n \rightarrow \infty$.

We now present a novel online algorithm, VIRTUAL+, that retains the simple structure of VIRTUAL and OPTIMISTIC, with no extra hyperparameters, but leads to a new state-of-the-art competitive ratio for $k < 5$. Our key insight is derived from re-examining the selection condition in the VIRTUAL algorithm and noticing that it is overly conservative and can be simplified. The VIRTUAL+ algorithm is presented in Algorithm 1, where the removed condition in VIRTUAL (L2-3) is ~~in pink strikethrough~~. Concretely, the condition that is used by VIRTUAL but *not* by VIRTUAL+ updates $R$ during the

selection phase without actually picking the item as part of $S_\mathcal{A}$. Essentially, this condition is theoretically convenient and leads to a simpler analysis by ensuring that the VIRTUAL algorithm never exceeds $k$ selections in $S_\mathcal{A}$. VIRTUAL+ removes this conservative $R$ update criteria in favor of a simple to implement condition, $|S_\mathcal{A}| \leq k$ line 4 (in pink). Furthermore, the new selection rule also retains the simplicity of VIRTUAL leading to a painless application to online attack problems.

---

**Algorithm 1** VIRTUAL and VIRTUAL+

**Inputs:** $t \in [k \dots n-k]$, $R = \emptyset$, $S_\mathcal{A} = \emptyset$

**Sampling phase:** Observe the first $t$ data points and construct a sorted list $R$ with the indices of the top $k$ data points seen. The method sort ensures: $\mathcal{V}(R[1]) \geq \mathcal{V}(R[2]) \cdots \geq \mathcal{V}(R[k])$.

**Selection phase**:{//VIRT+ removes L2-3 and adds L4 }

1: **for** $i := t+1$ to $n$ **do**
2: ~~**if** $\mathcal{V}(i) \geq \mathcal{V}(R[k])$ and $R[k] > t$ **then**~~
3: ~~$R = \text{sort}(R \cup \{i\} \setminus \{R[k]\})$~~
4: **else if** $\mathcal{V}(i) \geq \mathcal{V}(R[k])$ and $|S_\mathcal{A}| \leq k$ **then**
5: $R = \text{sort}(R \cup \{i\} \setminus \{R[k]\})$ {// Update $R$}

6: $S_\mathcal{A} = S_\mathcal{A} \cup \{i\}$ {// Select element $i$}

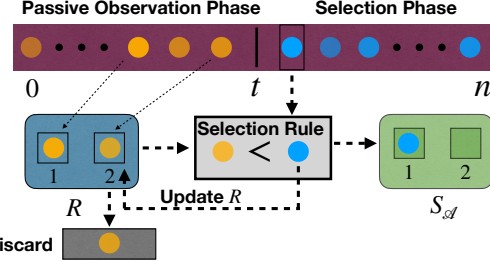

Figure 2: VIRTUAL+ observes $v_i$ (or estimates) and maintains $R$ during the sampling phase. Items are then picked into $S_\mathcal{A}$, after threshold $t$,

---

**Competitive ratio of VIRTUAL+.** What appears to be a minor modification in VIRTUAL+ compared to VIRTUAL leads to a significantly more involved analysis but a larger competitive ratio. In Theorem 1, we derive the analytic expression that is a tight lower bound for the competitive ratio of VIRTUAL+ for *general-k*. We see that VIRTUAL+ provably improves in competitive ratio for $k < 5$ over both VIRTUAL, OPTIMISTIC, and in particular the previous best single threshold algorithm, SINGLE-REF.

**Theorem 1.** *The competitive ratio of* VIRTUAL+ *for $k \geq 2$ with threshold $t_k = \alpha n$ can asymptotically be lower bounded by the following concave optimization problem,*

$$C_k \geq \max_{\alpha \in [0,1]} f(\alpha) := \alpha^k \sum_{m=0}^{k-1} a_m \ln^m(\alpha) - \alpha a_0 \quad where \quad a_m := \left( \frac{k^k}{(k-1)^{k-m}} - k^m \right) \frac{(-1)^{m+1}}{m!}.$$
(4)

*Particularly, we get $C_2 \geq 0.427, C_3 \geq .457, C_4 \geq .4769$ outperforming Albers & Ladewig (2020).*

**Connection to Prior Work**. The full proof for Theorem 1 can be found in §B along with a simple but illustrative proof for $k = 2$ in §A. Theorem 1 gives a tractable way to compute the competitive ratio of VIRTUAL+ for any $k$, that improve the previous state-of-the-art (Albers & Ladewig, 2020) in terms of single threshold $k$-secretary algorithms for $k < 5$ and $k > 100$.[1] However, it is also important to contextualize VIRTUAL+ against recent theoretical advances in this space. Most prominently, Buchbinder et al. (2014) proved that the $k$-secretary problem can be solved *optimally* (in terms of competitive ratio) using linear programs (LPs), *assuming a fixed length of $n$*. But these optimal algorithms are typically not feasible in practice. Critically, they require individually tuning multiple thresholds by solving a separate LP with $\Omega(nk^2)$ parameters for each length of the data stream $n$, and the number of constraints grows to infinity as $n \to \infty$. Chan et al. (2014) showed that optimal algorithms with $k^2$ thresholds could be obtained using infinite LPs and derived an optimal algorithm for $k = 2$ Nevertheless they require a large number of parameters and the scalability of infinite LPs for $k > 2$ remains uncertain. In this work, we focus on practical methods with a *single* threshold (i.e., with $O(1)$ parameters, e.g. Algorithm 1) that do not require involved computations that grow with $n$.

**Open Questions for Single Threshold Secretary Algorithms**. Albers & Ladewig (2020) proposed new non-asymptotic results on the $k$-secretary problem that outperform asymptotically optimal algorithms—opening a new range of open questions for the $k$-secretary problem. While this problem is considered solved when working with probabilistic algorithms[2] with $\Theta(nK^2)$ parameters (Buchbinder et al., 2014), finding optimal non-asymptotic single-threshold ($O(1)$ parameters) algorithms

---

[1]Albers & Ladewig (2020) only provide competitive ratios of SINGLE-REF for $k \leq 100$ and conclude that "a closed formula for the competitive ratio for any value of $k$ is one direction of future work". We partially answer this open question by expressing VIRTUAL+'s optimal threshold $t_k$ as the solution of a uni-dimensional concave optimization problem. In Table 3, we provide this threshold for a wide range of $k \geq 100$.

[2]At each timestep a deterministic algorithm chooses a candidate according to a deterministic rule depending on some parameters (usually a threshold and potentially a rank to compare with). A probabilistic algorithm

is still an open question. As a step towards answering this question, our work proposes a practical algorithm that improves upon Albers & Ladewig (2020) for $k = 2, \ldots, 4$ with an optimal threshold that can be computed easily as it has a closed form.

## 4 STOCHASTIC SECRETARY PROBLEM

In practice, online adversaries are unlikely to have access to the target model $f_t$. Instead, it is reasonable to assume that they have partial knowledge.

Following Papernot et al. (2017); Bose et al. (2020) we focus on modeling that partial knowledge by equipping the adversary with a surrogate model or representative classifier $f_s$. Using $f_s$ as opposed to $f_t$ means that we can compute the value $\mathcal{V}_i := \ell(f_s(x_i'), y_i)$ of an incoming data point. This value $\mathcal{V}_i$ acts as an estimate of the value of interest $v_i := \ell(f_t(x_i'), y_i)$. The *stochastic k-secretary problem* is then to pick, under the noise model induced by using $f_s$, the optimal subset $S_{\mathcal{A}}$ of size $k$ from $\mathcal{D}$. Thus, with no further assumptions on $f_s$ it is unclear whether online algorithms, as defined in §3.2, are still serviceable under uncertainty.

**Sources of randomness**. Our method relies on the idea that we can use the surrogate model $f_s$ to estimate the value of some adversarial examples on the target model $f_t$. We justify here how partial knowledge on $f_t$ could provide us an estimate of $v_i$. For example, we may know the general architecture and training procedure of $f_t$, but there will be inherent randomness in the optimization (e.g., due to initialization or data sampling), making it impossible to perfectly replicate $f_t$.

Moreover, it has been observed that, in practice, adversarial examples *transfer* across models (Papernot et al., 2016; Tramèr et al., 2017). In that context, it is reasonable to assume that the random variable $\mathcal{V}_i := \ell(f_s(x_i'), y_i)$ is likely to be close to $v_i := \ell(f_t(x_i'), y_i)$. We formalize this idea in Assumption 1

### 4.1 STOCHASTIC SECRETARY ALGORITHMS

In the stochastic $k$-secretary problem, we assume access to random variables $\mathcal{V}_i$ and that $v_i$ are fixed for $i = 1, \ldots, n$ and the goal is to maximize a notion of stochastic competitive ratio. This notion is similar to the standard competitive ratio defined in Eq. 2 with a minor difference that in the stochastic case, the algorithm does not have access to the values $v_i$ but to $\mathcal{V}_i$ that is an estimate of $v_i$. An algorithm is said to be $C_s$-competitive in the stochastic setting if asymptotically in $n$,

$$\mathbb{E}_{\pi \sim \mathcal{S}_n}[\mathbb{V}(S_{\mathcal{A}})] \geq (C_s + o(1))\mathbb{V}(S^*).$$

Here the expectation is taken over $\mathcal{S}_n$ (uniformly random permutations of the datastream $\mathcal{D}$ of size $n$) and over the randomness of $\mathcal{V}_i$, $i = 1, \ldots, n$. $S_{\mathcal{A}}$ and $S^*$ are the set of items chosen by the stochastic online and offline algorithms respectively (note that while the online algorithm has access to $\mathcal{V}_i$, the offline algorithm picks the best $v_i$) and $\mathbb{V}$ is a set-value function as defined previously.

**Analysis of algorithms**. In the stochastic setting, all online algorithms observe $\mathcal{V}_i$ that is an estimate of the actual value $v_i$. Since the goal of the algorithm is to select the $k$-largest values by only observing random variables $(\mathcal{V}_i)$ it is requisite to make a feasibility assumption on the relationship between values $v_i$ and $\mathcal{V}_i$. Let us denote $\text{top}_k\{v_i\}$ as the set of top-$k$ values among $(v_i)$.

**Assumption 1** (Feasibility). $\exists \gamma > 0$ *such that* $\mathbb{P}[\mathcal{V}_i \in \text{top}_k\{\mathcal{V}_i\} \mid v_i \in \text{top}_k\{v_i\}] \geq \gamma, \forall n \geq 0.$[3]

Assumption 1 is a feasibility assumption as if the ordering of $(\mathcal{V}_i)$ does not correspond at all with the ordering of $(v_i)$ then there is no hope any algorithm—online or an offline oracle—would perform better than random when picking $k$ largest $v_i$ by only observing $(\mathcal{V}_i)$. In the context of adversarial attacks, such an assumption is quite reasonable as in practice there is strong empirical evidence between the transfer of adversarial examples between surrogate and target models (see §C.1, for the empirical caliber of assumption 1). We can bound the competitive ratio in the stochastic setting.

**Theorem 2.** *Let us assume that* VIRTUAL+ *observes independent random variables* $\mathcal{V}_i$ *following Assumption 1. Its stochastic competitive ratio* $C_s$ *can be bounded as follows,*

$$C \geq C_s \geq \gamma C \tag{5}$$

---

choose to accept a candidate according to $q_{i,j,l}$ the probability of accepting the candidate in $i$-th position as the $j^{th}$ accepted candidate given that the candidate is the $l$-th best candidate among the $i$ first candidates ($i \in [n]$, $j, l \in [K]$.) See (Buchbinder et al., 2014) for more details on probabilistic secretary algorithms.

[3]Note that, for the sake of simplicity, the constant $\gamma$ is assumed to be independent of $n$ but a similar non-asymptotic analysis could be performed by considering a non-asymptotic definition of the competitive ratio.

The proof of Thm. 2 can be found in §C. Such a theoretical result is quite interesting as the stochastic setting initially appears significantly more challenging due to the non-zero probability that the observed ordering of historical values, $\mathcal{V}_i$, not being faithful to the true ranking based on $v_i$.

## 4.2 RESULTS ON SYNTHETIC DATA

We assess the performance of classical single threshold online algorithms and VIRTUAL+ in solving the stochastic $k$-secretary problem on a synthetic dataset of size $n = 100$ with $k \in [1, 10]$. The value of a data point is its index in $\mathcal{D}$ prior to applying any permutation $\pi \sim \mathcal{S}_n$ plus noise $\mathcal{N}(0, \sigma^2)$. We compute and plot the competitive ratio over $10k$ unique permutations of each algorithm in Figure 3.

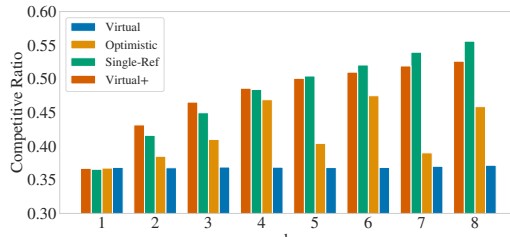

Figure 3: Estimation of the competitive ratio of online algorithms in the stochastic $k$-secretary problem with $\sigma^2 = 10$.

---

**Algorithm 2** Online Adversarial Attack

**Inputs:** Permuted Datastream: $\mathcal{D}_\pi$, Online Algorithm: $\mathcal{A}$, Surrogate classifier: $f_s$, Target classifier: $f_t$, Attack method: ATT, Loss: $\ell$, Budget: $k$, Online Fool rate: $F_\pi^{\mathcal{A}} = 0$.

1: **for** $(x_i, y_i)$ in $\mathcal{D}_\pi$ **do**
2: $\quad x_i' \leftarrow \text{ATT}(x_i)$ {// Compute the attack}
3: $\quad \mathcal{V}_i \leftarrow \ell(f_s(x_i'), y_i)$ {// Estimate $v_i$}
4: $\quad$ **if** $\mathcal{A}(\mathcal{V}_1, \dots, \mathcal{V}_i, k) == \text{TRUE}$ **then**
5: $\qquad F_\pi^{\mathcal{A}} \leftarrow F_\pi^{\mathcal{A}} + \frac{1\{f_t(x_i') \neq y_i\}}{k}$ {// Submit $x_i'$}
6: **return:** $F_\pi^{\mathcal{A}}$ {//$\mathcal{A}$ always submits $k$ attacks}

---

As illustrated for $k = 1$ all algorithms achieve the optimal $(1/e)$-deterministic competitive ratio in the stochastic setting. Note that the noise level, $\sigma^2$, appears to have a small impact on the performance of the algorithms (§E.2). This substantiates our result in Thm. 2 indicating that $C_n$-competitive algorithms only degrade by a small factor in the stochastic setting. For $k < 5$, VIRTUAL+ achieves the best competitive ratio—empirically validating Thm 1—after which SINGLE-REF is superior.

## 5 EXPERIMENTS

We investigate the feasibility of online adversarial attacks by considering an online version of the challenging NoBox setting (Bose et al., 2020) in which the adversary must generate attacks without any access, including queries, to the target model $f_t$. Instead, the adversary only has access to a surrogate $f_s$ which is similar to $f_t$. In particular, we pick at random a $f_t$ and $f_s$ from an ensemble of pre-trained models from various canonical architectures. We perform experiments on the MNIST LeCun & Cortes (2010) and CIFAR-10 Krizhevsky (2009) datasets where we simulate a $\mathcal{D}$ by generating 1000 permutations of the test set and feeding each instantiation to Alg. 2. In practice, online adversaries compute the value $\mathcal{V}_i = \ell(f_s(x_i'), y_i)$ of each data point in $\mathcal{D}$ by attacking $f_s$ using their fixed attack strategy (where $\ell$ is the cross-entropy), but the decision to submit the attack to $f_t$ is done using an online algorithm $\mathcal{A}$ (see Alg. 2). As representative attack strategies, we use the well-known FGSM attack (Goodfellow et al., 2015) and a universal whitebox attack in PGD (Madry et al., 2017). We are most interested in evaluating the online fool rate, which is simply the ratio of successfully executed attacks against $f_t$ out of a possible of $k$ attacks selected by $\mathcal{A}$. The architectures used for $f_s$, $f_t$, and additional metrics (e.g. competitive ratios) can be found in §E [4].

**Baselines**. We rely on two main baselines, first we use a NAIVE baseline–a lower bound–where the data points are picked uniformly at random, and an upper bound with the OPT baseline where attacks, while crafted using $f_s$, are submitted by using the true value $v_i$ and thus utilizing $f_t$.

**Q1: Utility of using an online algorithm**. We first investigate the utility of using an online algorithm, $\mathcal{A}$, in selecting data points to attack in comparison to the NAIVE baseline. For a given permutation $\pi$ and an attack method (FGSM or PGD), we compute the online fool rate of the NAIVE baseline and an $\mathcal{A}$ as $F_\pi^{\text{NAIVE}}$, $F_\pi^{\mathcal{A}}$ respectively. In Fig. 4, we uniformly sample 20 permutations $\pi_i \sim \mathcal{S}_n$, $i \in [n]$, of $\mathcal{D}$ and plot a scatter graph of points with coordinates $(F_{\pi_i}^{\text{NAIVE}}, F_{\pi_i}^{\mathcal{A}})$, for different $\mathcal{A}$'s, attacks with $k = 1000$, and datasets. The line $y = x$ corresponds to the NAIVE baseline performance —i.e. coordinates $(F_\pi^{\text{NAIVE}}, F_\pi^{\text{NAIVE}})$—and each point above that line corresponds to an $\mathcal{A}$ that outperforms

---

[4]Code can be found at: https://github.com/facebookresearch/OnlineAttacks

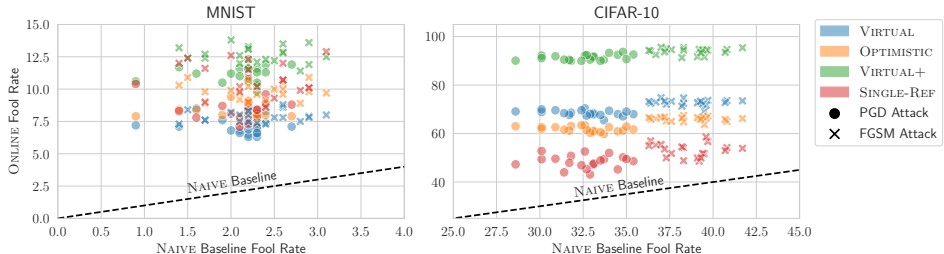

Figure 4: Plot of online fool rates for $k = 1000$ against PGD-robust models using different online algorithms $\mathcal{A}$, attacks, datasets, and 20 different permutations. For a given $x$-coordinate, a higher $y$-coordinate is better.

the baseline on a given $\pi_i$. As observed, all $\mathcal{A}$'s significantly outperform the NAIVE baseline with an average aggregate improvement of 7.5% and 34.1% on MNIST and CIFAR-10.

| | Algorithm | MNIST (Online fool rate in %) | | | CIFAR-10 (Online fool rate in %) | | | Imagenet (Online fool rate in %) | | |
|---|---|---|---|---|---|---|---|---|---|---|
| | | $k = 10$ | $k = 10^2$ | $k = 10^3$ | $k = 10$ | $k = 10^2$ | $k = 10^3$ | $k = 10$ | $k = 10^2$ | $k = 10^3$ |
| | NAIVE | 64.1 | 47.8 | 45.7 | 60.7 | 59.2 | 59.2 | 66.0 | 66.3 | 65.0 |
| | OPT | **87.0** | **84.7** | **83.6** | **86.6** | **87.3** | **86.5** | **98.7** | **95.3** | **96.2** |
| FGSM | OPTIMISTIC | 79.0 | 77.6 | 75.3 | 75.3 | 72.8 | 71.9 | 86.0 | 80.4 | 79.9 |
| | VIRTUAL | 78.6 | 79.1 | 77.4 | 76.1 | 77.1 | 75.4 | 85.3 | 84.9 | 84.3 |
| | SINGLE-REF | 85.1 | 83.0* | 72.3 | 80.4 | 84.0 | 66.0 | 94.0* | 92.4* | 72.5 |
| | VIRTUAL+ | 80.4 | 82.5* | 82.9 | 82.9 | 86.3 | 85.2 | 96.0* | 95.0* | 95.8 |
| | NAIVE | 69.7 | 67.2 | 67.9 | 72.5 | 70.4 | 68.6 | 72.5 | 72.5 | 73.8 |
| | OPT | **73.6** | **49.8** | **49.6** | **83.7** | **80.6** | **79.9** | **82.5** | **80.2** | **76.8** |
| PGD | OPTIMISTIC | 66.2 | 48.2 | 45.1 | 79.1 | 76.6 | 76.0 | 87.5* | 78.0* | 74.5* |
| | VIRTUAL | 63.4 | 46.2 | 46.8 | 78.3 | 77.5 | 76.9 | 80.0* | 74.0* | 75.6* |
| | SINGLE-REF | 71.5 | 49.7* | 42.9 | 80.2* | 79.6* | 74.5 | 77.5* | 79.5* | 75.2* |
| | VIRTUAL+ | 68.2 | 49.3* | 49.7 | 81.2* | 80.1* | 79.5 | 77.5* | 79.0* | 76.4* |

Table 1: Online fool rate of various online algorithms on non-robust models. For a given attack and value of $k$: ● at least 97%, ● at least 95%, ● at least 90%, ● less than 90% of the optimal performance. * indicates when there is several best methods with overlapping error bars. Detailed results with error bars can be found in §E.1.

**Q2: Online Attacks on Non-Robust Classifiers**. We now conduct experiments on non-robust MNIST, CIFAR-10, and Imagenet classifiers. We report the average performance of all online algorithms, and the optimal offline algorithm OPT in Tab. 5. For MNIST, we find that the two best online algorithms are SINGLE-REF and our proposed VIRTUAL+ which approach the upper bound provided by OPT. For experiments with $k < 5$ please see §E.5. For $k = 10$ and $k = 100$, SINGLE-REF is slightly superior while for $k = 1000$ VIRTUAL+ is the best method with an average relative improvement of 15.3%. This is unsurprising as VIRTUAL+ does not have any additional hyperparameters unlike SINGLE-REF which appears more sensitive to the choice of optimal thresholds and reference ranks, both of which are unknown beyond $k = 100$ and non-trivial to find in closed form (see §E.3 for details). On CIFAR-10, we observe that VIRTUAL+ is the best approach regardless of attack strategy and the online attack budget $k$. Finally, for ImageNet we find that all online algorithms improve over the NAIVE baseline and approach saturation to the optimal offline algorithm, and as a result, all algorithms are equally performant—i.e. within error bars (see §E.1 for more details). A notable observation is that even conventional whitebox adversaries like FGSM and PGD become strong blackbox transfer attack strategies when using an appropriate $\mathcal{A}$.

**Q3: Online Attacks on Robust Classifiers**. We now test the feasibility of online attacks against classifiers robustified using adversarial training by adapting the public Madry Challenge (Madry et al., 2017) to the online setting. We report the average performance of each $\mathcal{A}$ in Table **??**. We observe that VIRTUAL+ is the best online algorithms, outperforming VIRTUAL and OPTIMISTIC, in all settings except for $k = 10$ on MNIST where SINGLE-REF is slightly better.

**Q4: Differences between the online and offline setting**. The online threat model presents several interesting phenomena that we now highlight. First, we observe that a stronger attack (e.g. PGD)—in comparison to FGSM—in the offline setting doesn't necessarily translate to an equivalently stronger attack in the online setting. Such an observation was first made in the conventional offline transfer setting by Madry et al. (2017), but we argue the online setting further exacerbates this phenomenon. We explain this phenomenon in Fig. 5a & 5b by plotting the ratio of unsuccessful attacks to total attacks as a function of loss values for PGD and FGSM. We see that for the PGD attack numerous

| | Algorithm | MNIST (Online fool rate in %) | | | CIFAR-10 (Online fool rate in %) | | |
|---|---|---|---|---|---|---|---|
| | | $k = 10$ | $k = 100$ | $k = 1000$ | $k = 10$ | $k = 100$ | $k = 1000$ |
| FGSM | NAIVE | $2.1 \pm 4.5$ | $2.1 \pm 1.4$ | $2.1 \pm 0.4$ | $31.9 \pm 14.2$ | $32.6 \pm 4.7$ | $32.5 \pm 1.5$ |
| | OPT | $\mathbf{80.0 \pm 0.0}$ | $\mathbf{55.0 \pm 0.0}$ | $\mathbf{18.9 \pm 0.0}$ | $\mathbf{100.0 \pm 0.0}$ | $\mathbf{100.0 \pm 0.0}$ | $\mathbf{97.2 \pm 0.0}$ |
| | OPTIMISTIC | $49.7 \pm 0.6$ | $25.7 \pm 0.1$ | $9.7 \pm 0.0$ | $72.4 \pm 0.5$ | $64.6 \pm 0.1$ | $61.9 \pm 0.0$ |
| | VIRTUAL | $49.8 \pm 0.5$ | $27.8 \pm 0.1$ | $8.1 \pm 0.0$ | $75.1 \pm 0.5$ | $74.3 \pm 0.1$ | $68.9 \pm 0.0$ |
| | SINGLE-REF | $62.0 \pm 0.7$ | $45.2 \pm 0.2$ | $10.2 \pm 0.0$ | $84.3 \pm 0.6$ | $90.9 \pm 0.3$ | $48.6 \pm 0.1$ |
| | VIRTUAL+ | $68.2 \pm 0.5$ | $42.2 \pm 0.1$ | $12.7 \pm 0.0$ | $91.5 \pm 0.4$ | $96.5 \pm 0.1$ | $91.7 \pm 0.0$ |
| PGD | NAIVE | $1.8 \pm 4.1$ | $1.9 \pm 1.4$ | $1.9 \pm 0.4$ | $39.1 \pm 14.2$ | $38.9 \pm 4.4$ | $38.7 \pm 1.5$ |
| | OPT | $\mathbf{58.9 \pm 0.4}$ | $\mathbf{39.9 \pm 0.1}$ | $\mathbf{16.1 \pm 0.0}$ | $\mathbf{100.0 \pm 0.0}$ | $\mathbf{100.0 \pm 0.0}$ | $\mathbf{98.0 \pm 0.0}$ |
| | OPTIMISTIC | $34.9 \pm 0.5$ | $19.2 \pm 0.1$ | $8.2 \pm 0.0$ | $75.4 \pm 1.9$ | $68.5 \pm 0.4$ | $66.0 \pm 0.1$ |
| | VIRTUAL | $35.4 \pm 0.5$ | $21.8 \pm 0.1$ | $7.2 \pm 0.0$ | $78.1 \pm 1.7$ | $77.3 \pm 0.5$ | $72.8 \pm 0.1$ |
| | SINGLE-REF | $44.1 \pm 0.6$ | $33.9 \pm 0.2$ | $8.3 \pm 0.0$ | $86.2 \pm 2.2$ | $91.9 \pm 0.9$ | $53.2 \pm 0.3$ |
| | VIRTUAL+ | $48.3 \pm 0.5$ | $32.8 \pm 0.1$ | $11.1 \pm 0.0$ | $92.2 \pm 1.3$ | $97.1 \pm 0.4$ | $94.2 \pm 0.1$ |

Table 2: Online fool rate of various online algorithms on robust models. For a given attack and value of $k$: • at least 90%, • at least 80%, • at least 70%, • less than 70% of the optimal performance.

unsuccessful attacks can be found even for high surrogate loss values and as a result, can lead $\mathcal{A}$ further astray by picking unsuccessful data points—which may be top-$k$ in surrogate loss values—to conduct a transfer attack. A similar counter-intuitive observation can be made when comparing the online fool rate on robust and non-robust classifiers. While it is natural to expect the online fool rate to be lower on robust models we empirically observe the opposite in Tab. 5 and **??**. To understand this phenomenon we plot the ratio of unsuccessful attacks to total attacks as a function $f_s$'s loss in Fig. 5c and observe non-robust models provide a non-vanishing ratio of unsuccessful attacks for large values of $\mathcal{V}_i$ making it harder for $\mathcal{A}$ to pick successful attacks purely based on loss (see also §F).

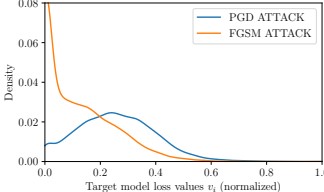

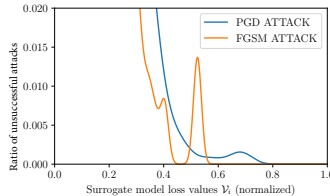

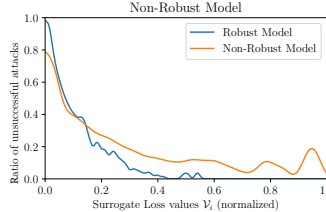

(a) Distribution of $f_t$'s loss values.  (b) Ratio of unsuccessful attacks.  (c) Ratio of unsuccessful attacks.

Figure 5: For every example in MNIST we compute an attack using $f_s$ and submit it to $f_t$. **Left**: The distribution of the normalized loss values of $f_t$ for all attacks where a higher loss is a stronger attack. **Middle**: The percentage of unsuccessful attacks as a function of $f_s$ normalized loss values. **Right**: smoothed ratio of unsuccessful attacks to total attacks as a function of the $f_s$ normalized loss values.

## 6 CONCLUSION

In this paper, we formulate the online adversarial attack problem, a novel threat model to study adversarial attacks on streaming data. We propose VIRTUAL+, a simple yet practical online algorithm that enables attackers to select easy to fool data points while being theoretically the best single threshold algorithm for $k < 5$. We further introduce the stochastic $k$-secretary problem and prove fundamental results on the competitive ratio of any online algorithm operating in this new setting. Our work sheds light on the tight coupling between optimally selecting data points using an online algorithm and the final attack success rate, enabling weak adversaries to perform on par with stronger ones at no additional cost. Investigating, the optimal threshold values for larger values of $k$ along with competitive analysis for the general setting is a natural direction for future work.

## ETHICS STATEMENT

We introduce the online threat model which aims to capture a new domain for adversarial attack research against streaming data. Such a threat model exposes several new security and privacy risks. For example, using online algorithms, adversaries may now tailor their attack strategy to attacking a small subset of streamed data but still cause significant damage to downstream models e.g. the control system of an autonomous car. On the other hand our research also highlights the need and importance

of stateful defence strategies that are capable of mitigating such online attacks. On the theoretical side the development and analysis of VIRTUAL+ has many potential applications outside of adversarial attacks broadly categorized as resource allocation problems. As a concrete example one can consider advertising auctions which provide the main source of monetization for a variety of internet services including search engines, blogs, and social networking sites. Such a scenario is amenable to being modelled as a secretary problem as an advertiser may be able to estimate accurately the bid required to win a particular auction, but may not be privy to the trade off for future auctions.

## REPRODUCIBILITY STATEMENT

Throughout the paper we tried to provide as many details as possible in order for the results of the paper to be reproducible. In particular, we provide a detailed description of VIRTUAL+in Alg. 1 and we explain how to combine any attacker (e.g. PGD) with an online algorithm to form an online adversarial attack in Alg. 2. We provide a general description of the experimental setup in §5, further details with the specific architecture of the models and hyper-parameters used are provided in §E.3. We also provided confidence intervals with our experiments every time it was possible to do so. Finally the code used to produce the experimental results is provided with the supplementary materials and will be made public after the review process.

## ACKNOWLEDGEMENTS

The authors would like to acknowledge Manuella Girotti, Pouya Bashivan, Reyhane Askari Hemmat, Tiago Salvador and Noah Marshall for reviewing early drafts of this work.

**Funding.** This work is partially supported by the Canada CIFAR AI Chair Program (held at Mila). Joey Bose was also supported by an IVADO PhD fellowship. Simon Lacoste-Julien and Pascal Vincent are CIFAR Associate Fellows in the Learning in Machines & Brains program. Finally, we thank Facebook for access to computational resources.

## CONTRIBUTIONS

*Andjela Mladenovic* and *Gauthier Gidel* formulated the online adversarial attacks setting by drawing parallels to the $k$-secretary problem, with *Andjela Mladenovic* leading the theoretical investigation and theoretical results including the competitive analysis for VIRTUAL+ for the general-$k$ setting. *Avishek Joey Bose* conceived the idea of online attacks, drove the writing of the paper and helped *Andjela Mladenovic* with experimental results on synthetic data. *Hugo Berard* was the chief architect behind all experimental results on MNIST and CIFAR-10. *William L. Hamilton*, *Simon Lacoste-Julien* and *Pascal Vincent* provided feedback and guidance over this research while *Gauthier Gidel* supervised the core technical execution of the theory.

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
