# OpenReview forum: "Online Adversarial Attacks"
_ICLR.cc/2022/Conference — ICLR 2022 Poster_

### Official Review · Reviewer_XV1X · 2021-10-24

**Correctness:** 3
**Technical Novelty And Significance:** 3
**Empirical Novelty And Significance:** 3
**Recommendation:** 5
**Confidence:** 3

**Main Review:**

The new algorithm VIRTUAL+ is interesting, but I vote for weak reject given its motivation and novelty towards adversarial learning. The explanations towards some observations in the experiments do not convince me either.

(1) It is not clear why this paper considers the online adversarial attack. The main contributions are built upon the proposed new algorithm towards k-secretary problem, and the online adversarial attack is only one of the possible applications of this k-secretary problem. Different from most literature in adversarial training, it does not deepen the understanding of attack method or robust features. The current title and abstract are somewhat not accurate to describe the main interest of this paper. Please consider polish the writing.

(2) Given the problem setup of "selecting k samples to attack in a stream of data to mislead a fixed neural network to make wrong predictions", the proposed problem forumlation in this paper is a natural solution. However, in reality, e.g. filtering system to detect spam email, the classifier is updated along the time. It is more interesting to consider a changing f_t rather than a fixed target f_t. Similarly, we can also train f_s through receiving the streaming data.

(3) In the experiment, the authors observe a much-better-than-PGD performance for FSGM. In theory a PGD attack is always not weaker than FSGM. I checked the code and found that the PGD attack setup is (nb_iter=40,eps_iter=0.01) for eps=0.3 in MNIST. This setup is used in other literature, but could you please try to increase nb_iter to see its performance? It is possible that the attack is hard to train for some data, so a larger nb_iter may help diagnose this. The current explanation for this observation does not convince me.

(4) In the experiment, the authors observe that in CIFAR-10, the robust model is more vulnerable to online attack compared to the non-robust model. To explain this, the authors mention that in Fig. 5c the non-robust models have a heavier tail making it harder for A to pick successful attacks. I do not understand this explanation. If the non-robust models have a heavier tail, does this mean that more data are easy to attack in the non-robust model? Or do you mean that the loss does not reflect the exact decision? Since this observation is counter-intuitive, please have some detailed and convincing explanation for it.


Minor things:

(5) The paper provides motivation and intuition of the proposed VIRTUAL+ algorithm. It would be great if discussions can be provided for the technical difficulties of Theorem 1.

(6) The title "Non-Robust Model" in Figure 5(c) is misleading. Please remove it. Also the caption of Figure 5 says "MNIST (or CIFAR)", please make it clear which dataset the subfigures are using.


**Summary Of The Paper:**

This paper formulates an online threat model and use the VIRTUAL+ algorithm their proposed to solve this k-secretary problem. The proposed method is supported by theoretical justifications, and the numerical experiment shows a better performance of VITRUAL+ over other existing algorithms.

**Summary Of The Review:**

The new algorithm VIRTUAL+ is interesting, but please make it clear about the motivation and novelty of this work towards adversarial learning. Please also provide convincing explanations towards some observations in the experiments.

---

> ### Author Response · Authors · 2021-11-17
> **Response to Reviewer XV1X part 1/2**
>
> We would like to thank the reviewer for appreciating our contribution regarding Virtual+ algorithm. Furthermore, we respect the constructive skepticism of Reviewer XV1X.
> We think it illustrates well the fact that non-trivial things happen when considering the online adversarial attack setting. We hope that this discussion and our additional experiments will convince them that our work is of interest to the ICLR community by its results and the insights it brings. We now develop on these points:
>
> **Point (1)**
> “It is not clear why this paper considers the online adversarial attack”:
> A: We have several contributions in this paper. One indeed regard Virtual+ algorithm, but the aspect of merging ideas from two fields is also part of the contributions of this papers:
> - We combine ideas from the secretary problems and Adversarial examples to study the problem of adversarial attacks in an online fashion.
> - We experimentally show that our combination of online algorithms and adversarial attacks using a surrogate classifier is drastically efficient to detect the most vulnerable examples in an online fashion
> - We discovered some interesting insights about robust and non-adversarially trained models as well as FSGM and PDG.  We will develop these points in the answers to (3) and (4).
>
> R: “The current title and abstract are somewhat not accurate to describe the main interest of this paper.”
>
> A: We are slightly confused by this statement. We develop a new threat model that precisely corresponds to online adversarial attacks. We introduce secretary algorithms as a tool to understand and solve this problem, but our final goal is to deal with online adversarial examples.
> Could Reviewer XV1X further develop this point or propose an example of a title that would describe better the main interest of this paper?
>
> “Please consider polish the writing.”
> A: We would like to mention that we are slightly surprised by this request as Reviewer 82UN mentioned that our paper “is in well structure and easy to follow”
> We agree there are some minor issues that we will fix (for instance, you comment (6) ), but we believe these details do not affect the overall quality of our writing.
>
> **Point (2)**
> “It is more interesting to consider a changing f_t rather than a fixed target f_t. Similarly, we can also train f_s through receiving the streaming data”
>
> A: We respectfully disagree with this statement. We think both settings are interesting.
> To motivate the setting where $f_t$ is not changing, we would like to point out that most machine learning models deployed in production are not updated while receiving data but are periodically retrained to deal with domain shift [1,2,3].  Also, we would like to point out that one of the most notorious examples of the dangers of ML language models was with a changing model that was learning with the streaming of data [4].
>
> **Point (3)**
> “It is possible that the attack is hard to train for some data, so a larger nb_iter may help diagnose this.”
> ”The current explanation for this observation does not convince me”
>
> A: To convince Reviewer XV1X we have two arguments to present that we think are strong.
> First, a rich body of literature already noted that iterated attacks lead to less transferable attacks [5,6,7]. Particularly [5] note that there is a “ popular belief that the white-box strength of an attack is at odds with its transferability”. Our online adversarial setting exacerbates this aspect since it aims at selecting the best surrogate value (and hoping that it will transfer). Secondly, we performed additional experiments with a larger number of iterations for PGD as the reviewer suggested but found no appreciable difference between the existing results in Table 1 & 2.
>
> **Point (4)**  “If the non-robust models have a heavier tail, does this mean that more data are easy to attack in the non-robust model?”
>
> A: In Fig 5c,  we plot the distribution of the *ratio* of successful vs. unsuccessful attacks (as mentioned in our caption). The fact that the non-robust (non-adversarially trained) models have a heavy tail and that robust models do not can has the following interpretation:
> - Large values for an attack on the surrogate classifier may not always transfer well to the target classifier
> - For robust models, there is a clear separation (in terms of surrogate loss value) between successful and unsuccessful attacks, meaning that the attacks with a large value for the surrogate classifier are very likely to transfer well.
>
> Response Part 1/2

---

> > ### Author Response · Authors · 2021-11-17
> > **Response to Reviewer XV1X Part 2/2**
> >
> > **Point 5:**
> > The paper provides motivation and intuition of the proposed VIRTUAL+ algorithm. It would be great if discussions can be provided for the technical difficulties of Theorem 1.
> >
> > A. The technical difficulty of achieving the competitive ratio of Virtual+ and proving the Theorem1 lies in the dynamic nature of Virtual+. That is, Virtual+ updates the reference set $R$ for every element that is selected which makes the proof challenging as the element that is used to compare when selecting items into our selection set may be updated due to $R$ itself being updated.. Furthermore, by optimizing for the competitive ratio and not simplicity of proof our condition that the number of elements picked before the selection has to be k, even further complicates the proof.
> > The selection criteria for our algorithm Virtual+ relies on two decoupled selection conditions which are
> > Element at time step $i$ has to be larger than the k-th element seen up to that point
> > The total number of selected elements is less than k before time step $i$
> > In previous algorithms, such as Virtual and Optimistic there was only one selection criteria which implied criteria 2 in Virtual+ automatically. In our algorithm, constructing these two conditions decoupled, allowed us to optimize for the competitive ratio at the expense of conducting more complex analysis. In the Single Ref algorithm, while these two conditions are decoupled, the selection criteria involves a comparison with the static element, and therefore in contrast to it, in Virtual+ algorithm the same proof techniques could not be reused because of the dynamic nature of selection criteria 1 in our algorithm.
> > To summarize, having two decoupled conditions with one of them being dynamically updated is the source of the major complexity in the analysis of our algorithm.
> >
> > **Point (6):**
> > Thank you for your suggestions. We have implemented them in the revision of our paper.
> >
> >
> > [1] Perdomo, Juan, et al. "Performative prediction." International Conference on Machine Learning. PMLR, 2020.
> >
> > [2] Gama, João, et al. "A survey on concept drift adaptation." ACM computing surveys (CSUR) 46.4 (2014): 1-37.
> >
> > [3] Žliobaitė, Indrė. "Learning under concept drift: an overview." arXiv preprint arXiv:1010.4784 (2010).
> >
> > [4] Schwartz, Oscar. "In 2016, Microsoft’s racist chatbot revealed the dangers of online conversation." IEEE Spectrum 11 (2019).
> >
> > [5] Alexey Kurakin, Ian Goodfellow, and Samy Bengio. Adversarial machine learning at scale. In ICLR, 2017.
> >
> > [6] Alexey Kurakin, Ian Goodfellow, and Samy Bengio. Adversarial examples in the physical world. ICLR2017 workshop,.
> >
> > [7] Yinpeng Dong, Fangzhou Liao, Tianyu Pang, Hang Su, Jun Zhu, Xiaolin Hu, and Jianguo Li. Boosting adversarial attacks with momentum. In CVPR, 2018.
> >
> > Response Part 2/2

---

> > > ### Author Response · Authors · 2021-11-23
> > > **Effect of number of iterations in PGD**
> > >
> > > To provide empirical evidence to our claim that increasing the number of iterations in PGD---i.e. nb_iter=40 -> nb_iter=100---does not increase online fool rate we conduct an experiment on Imagenet models. Please note, we have updated our manuscript with new results on Imagenet (Table 1 and Appendix E) and below we report results with the sole change to PGD being we set the hyper parameter nb_iter=100. We find that a stronger PGD attack has worse transfer properties than the ones with nb_iter=40 which we report in Table 1 and Appendix E. These results are unsurprising as there is empirical evidence in the community that stronger iterated attacks have worse transfer properties (see refs 5,6,7 above) and in the online setting this phenomena is exacerbated in our online threat model. We hope this fully addresses point 3 in the reviewers original review.
> > >
> > > K = 10
> > >
> > > OPT:  70.0 $\pm$ 8.2
> > >
> > > VIRTUAL: 73.3 $\pm$ 9.8
> > >
> > > OPTIMISTIC: 70.0 $\pm$ 12.5
> > >
> > > VIRTUAL+ : 66.7 $\pm$ 10.9
> > >
> > > SINGLE_REF: 76.7 $\pm$ 7.2
> > >
> > > NAIVE: 80.0 $\pm$ 0.0
> > >
> > > K = 100
> > >
> > > OPT: 66.5 $\pm$ 6.0
> > >
> > > VIRTUAL: 68.0 $\pm$ 7.1
> > >
> > > OPTIMISTIC: 68.0 $\pm$ 6.4
> > >
> > > VIRTUAL+ : 71.0 $\pm$ 4.9
> > >
> > > SINGLE_REF: 68.0 $\pm$ 7.8
> > >
> > > NAIVE: & 58.0 $\pm$ 0.0
> > >
> > > K = 1000
> > >
> > > OPT: 70.5 $\pm$ 3.1
> > >
> > > VIRTUAL:  71.4 $\pm$ 5.0
> > >
> > > OPTIMISTIC: & 70.5 $\pm$ 3.9
> > >
> > > VIRTUAL+ : 72.8 $\pm$ 4.3
> > >
> > > SINGLE_REF: 68.6 $\pm$ 3.7
> > >
> > > NAIVE: 65.7 $\pm$ 2.5

---

> > > > ### Comment · Reviewer_XV1X · 2021-11-23
> > > > **Thanks for the response**
> > > >
> > > > Thanks for running many additional experiments and updating the manuscript.
> > > >
> > > > I think it is a good contribution to introduce VIRTUAL+ in this paper, but I have concerns towards the part for the online adversarial attack:
> > > >
> > > > First, the authors ran a lot of experiments to verify the correctness of the observations, but the observation that robust model performs worse than non-robust model is very counter-intuitive, so it perhaps needs more detailed investigations. The counter-intuitive phenomenons themselves even worth a paper.
> > > >
> > > > Second, more evidences are essential to support the novelty and importance of the contribution of online adversarial attack. If the problem exists in real practice, why shall we consider the model in this paper, rather than consider the need (e.g. budget) from real practice? In addition, it would be great if the paper can highlight the uniqueness of the proposed solution.

---

> > > > > ### Author Response · Authors · 2021-11-24
> > > > > **Re:Thanks for the response**
> > > > >
> > > > > We thank the reviewer for responding to our rebuttal and acknowledging Virtual+ as a significant contribution to the current paper.
> > > > >
> > > > > **Regarding the counter-intuitive phenomenon that robust models perform worse than non-robust models.**
> > > > >
> > > > > We believe that we have a clear and natural explanation for this phenomenon that we supported with empirical evidence in Figures 5 and Appendix F”: it is because there is a clearer separation between unsuccessful and successful ***top-$k$*** transfer attacks for robust models than for non-robust models.
> > > > > Put another way, when the attacker achieves a high surrogate loss value (relative to other surrogate losses), the attack is more likely to succeed for robust models. Since the online algorithm picks only top-k, where $k << n$, it can achieve higher online fool rates by only selecting the few attacks with a high surrogate value. We hope this fully clarifies the counter-intuitive phenomena. If there is anything that remains unclear, we ask the reviewer for more guidance.
> > > > >
> > > > > We would also like to push back against the assertion that more evidence and novelty is needed for online adversarial attacks. We believe introducing a new threat model and theoretically showing the feasibility of online attacks in this setting already highlights critical attack vulnerabilities that are simply not present in the literature. As to the reviewer comment on picking an online attack budget $k$ from practice, we would like to remind the reviewer even the conventional $\epsilon$-budget in $l_{\infty}$-attacks (e.g,. 8/256) is a hyperparameter, not motivated from practice but generally agreed upon in the community for a unified study of attacks and defenses. In this vein, we may view $k$ as another analogous hyperparameter that the attacker may choose using their domain knowledge. More importantly, our theoretical analysis-- a primary consideration in this work-- informs the attacker how best to pick $k$.
> > > > >
> > > > >
> > > > > Finally, we stress that our entire contribution regarding online adversarial attacks, including the online threat model, an application of secretary algorithms to pick online attacks (Algorithm 2), and their empirical demonstration of performance, are all novel contributions.
> > > > > We detailed these contributions in our “Response to Reviewer XV1X part 1/2”. We think these contributions are unique. Can the reviewer point us to any work in the literature that we may not be aware of?

---

> ### Author Response · Authors · 2021-11-23
> **Kind Reminder to Respond to our Rebuttal**
>
> Dear Reviewer,
>
> We appreciate the time and energy you've dedicated to your review. We have updated our manuscript and provided a detailed response to the key points raised in our rebuttal below. We have also clarified the motivation for online attacks in a general response to all reviewers. We would greatly appreciate a response from you and in particular whether our rebuttal has successfully addressed all of your concerns, potentially allowing you to update your score. We are also equally happy to answer any further remaining questions that arise that may help make your evaluation of our work clearer.

---

> ### Author Response · Authors · 2021-11-30
> **Pleasant Reminder to respond to our Response**
>
> Dear Reviewer,
>
> We are fast approaching the end of the discussion period and we would like to thank you for responding to our original rebuttal. We have responded to your post-rebuttal comments and were wondering if these new comments have assuaged your concerns. Please note that Reviewers 82UN and VskD upgraded their scores from 6->8 and 5->6 respectively while sharing much of your initial concerns regarding the counter-intuitive phenomena and the motivation and impact of our online attacks setting. We hope our response to your post-rebuttal comments helped in answering any lingering questions and potentially allows you to endorse our paper more wholeheartedly.

---

### Official Review · Reviewer_4rG9 · 2021-10-30

**Correctness:** 2
**Technical Novelty And Significance:** 3
**Empirical Novelty And Significance:** 2
**Recommendation:** 5
**Confidence:** 4

**Main Review:**

Instead of studying how to craft an adversarial example x', this work supposes that the attacking strategy of finding x' is given and focuses on finding the right subset of data points to attack in an online fashion.

One question the audience may have is under what real-world circumstances, the attacks can be carried out in such an online fashion.

* The authors mentioned networked control systems. I agree that control systems operate in real-time. But it does not mean that the attack model here can be applied to attacks on control systems, because in control systems current state depends on previous states (so do the data points, which means data points are not independently ordered). Also, in control systems, usually, the goal is not to optimize certain loss functions but to ensure stability.

* Can the authors provide more real-world examples/situations where the attack model studied here can be applied? If possible, please consider conducting experiments on these examples.

* For MNIST and CIFAR-10, data points usually arrive in batches instead of single data points. Can the authors also justify that in the manuscript? How would the authors deal with data batches? Can the current model extend seamlessly to the "batched data" situation?

Another major concern is the performance of the VIRTUAL+ algorithm. The theoretical results show that the algorithm produces the best competitive ratio only when k<5. In real applications, the number of data points involved can reach millions if not billions. The number of data points being attacked can easily go beyond 5. Does it mean that the VIRTUAL+ algorithm can only perform relatively better under some extreme cases and the application of the algorithm is limited?

Other minor concerns include:
* In C.1, it is not clear how table 4 is produced. In particular, it is not clear how the surrogate model is generated.
* please mention several ways of counter-attack in the conclusion section, e.g., arranging data points in a way such that the effect of  VIRTUAL+ attacks is mitigated.
* Figure 4 appears before figure 3

**Summary Of The Paper:**

 This paper studies the online adversarial attacks on deep learning (DL) models. Unlike other works that focused on adversarial attacks on  DL models, this work studies, on half of the attacker, how to choose a subset of data points to attack in an online fashion. Since the data arrives as a stream, the attacker has to make an irrevocable decision on whether to attack when each data point comes. This setting aligns with the situation given in the well-studied k-secretary problem. Built on top of a famous algorithm in the k secretary problem called VIRTUAL, an algorithm called VIRTUAL+ is proposed by the authors, which provably generates the best competitive ratio among all the single threshold-based algorithms (when the number of data points being attacked is less than 5). The attacker may not have access to the target deep learning model but can estimate the query value. Hence, the authors obtain theoretical bounds on the performance of the proposed VIRTUAL+ algorithm for the stochastic setting. This paper conducts experiments on both MNIST and CIFAR-10 with both vanilla and robust classifiers.


**Summary Of The Review:**

In the review, I proposed two major concerns toward the paper along with some minor ones. I did not check the proof carefully but they look correct.

---

> ### Author Response · Authors · 2021-11-17
> **Response to Reviewer 4rG9**
>
> We thank the reviewer for the comments and feedback. We now answer the key points raised by the reviewer grouped by theme.
>
> **Motivation for Online Attacks:**
> Please see our general response to all reviewers above.
>
> **Batched Data**
> The reviewer asks an interesting question regarding online attacks on batch data. Unfortunately, a key aspect of the online threat model is that the data is streamed, and as such, a batched setting is simply not possible. Moreover, our proposed online threat model is best viewed as an inference/test time attack, and in accordance with the literature of conventional adversarial attacks, it is standard practice to attack a single data point rather than a batch for classical attacks like FGSM PGD, and its variants. To the best of our knowledge, numerous variants of black box threat models (e.g., query-based) consider attacking a single data point and refining the attack iteratively (see [2] for an example), and as such, data does not arrive in a batch. We kindly ask the reviewer to further clarify what they mean by batched data if we have misinterpreted their comment.
>
>
> **Performance of Virtual+ for k >5:**
> The reviewer makes a great observation that our theoretical results for the competitive ratio is for k < 5. While we agree that in many real world applications $k < 5$ might be artificial which is why in our experiments we consider $k=10,100,1000$ for both MNIST and CIFAR10 (see Tables 1 & 2). In these settings, we still find that all secretary algorithms perform significantly better than the Naive baseline and Virtual+ consistently performs the best, almost reaching the Optimal offline solution, which has oracle access. *Please note, the use of a secretary algorithm to submit online attacks---and not just solely Virtual+----is a contribution of our work. Thus the relative difference in performance between Naive and any other secretary algorithm is more significant than what might have initially come across. Therefore we would like to politely push back against the concern raised by the reviewer that Virtual+ is not performant beyond k > 5.
>
> In addition, please note one of the principle design choices of our proposed online threat model is that $k$ is chosen judiciously such that an attacker is forced to attack a small subset of the streamed dataset to avoid detection. Thus, increasing $k$ without care runs the risk of being detected by stateful and adaptive defenses [1] that are known to increase their efficacy as they observe more adversarial samples. We agree this measure may not precisely correspond to the probability of being detected. Still, we argue that it is a good approximation that can be universally applied to any online transfer attacks.
>
> **Minor Concerns**
> Q. “In C.1, it is not clear how table 4 is produced”
> The table considers the intersection of the top-k surrogate loss values that are top-k according to the true target model’s loss for a total of $n=10000$ data points. The exact strategy that we used to pick our surrogate models is outlined in Appendix E.2 and also in our response to Reviewer 82UN. In short, each surrogate and target model is picked without replacement from an ensemble of 5 model architectures that are trained offline.
> We understand that in C.1 this was not sufficiently clear and we will add a pointer to the Appendix E.2 for further clarification.
>
> Q. “please mention several ways of counter-attack in the conclusion section, e.g., arranging data points in a way such that the effect of VIRTUAL+ attacks is mitigated.”
>
> This is an interesting point. Under the standard definition of competitive ratio in the online algorithms literature, we consider an expectation over all permutations of the dataset. Thus our theory guarantees the performance of Virtual+ in this expectation. But, of course, if a second player had the power to select the permutation adversarially so that the top-k always occurred in the sampling phase, the effectiveness of Virtual+ attacks would be significantly diminished. We will update the conclusion to highlight this nuance but note that analysis of online algorithms in such adversarial settings represents an entirely different type of problem outside the scope of this work.
>
> Q. Fig 4 appears before Fig 3.
>
> Great catch! We have fixed this in the updated manuscript.
>
> [1]  “Stateful detection of black-box adversarial attacks” by Chen et al. SPAI 2020
>
> [2] Ilyas, Andrew, Logan Engstrom, and Aleksander Madry. "Prior convictions: Black-box adversarial attacks with bandits and priors." arXiv preprint arXiv:1807.07978 (2018).

---

> ### Author Response · Authors · 2021-11-23
> **Kind Reminder to Respond to our Rebuttal**
>
> Dear Reviewer,
>
> We thank you for the time and effort spent in reviewing our paper. We wish to inform you that we have updated our manuscript as well as responded to your main concerns in our rebuttal below. We have also written a general response to all reviewers where we highlight the potential real world settings where an adversary could launch online attacks We hope our responses were satisfactory in clarifying any remaining points that hinder you from potentially reconsidering your evaluation of our work. We are also happy to respond to any new guidance or questions that you may have.

---

> ### Author Response · Authors · 2021-11-29
> **Pleasant reminder to respond to our rebuttal**
>
> Dear Reviewer,
>
> We are nearing the end of the discussion period, so if we there are any lingering concerns that were not adequately addressed in our responses, we would love to hear soon so that it is possible for us to respond. As a quick summary we have clarified the motivation---with concrete real world examples---for online attacks in our general response to all reviewers. In addition, we have added new ImageNet results in Table 1 along with revisions to Appendix F and a new Appendix G. These new changes and our led to Reviewer 82UN upgrading their score from 6->8 and Reviewer VskD to go from 5->6. We were wondering if our rebuttal and updates to the manuscript resolved all of your two key concerns such that you could support this paper more fully. Thanks again for your time and consideration for our work.

---

> ### Author Response · Authors · 2021-11-29
> **Pleasant reminder**
>
> Dear Reviewer,
>
> We are nearing the end of the discussion period, so if we there are any lingering concerns that were not adequately addressed in our responses, we would love to hear soon so that it is possible for us to respond. Please note that Reviewer 82UN also shared similar concerns regarding the counter-intuitive phenomena we observe in the online setting but our responses and updates to the manuscript helped assuage these concerns and led to them increasing their score from 6->8. We thank you again for your time and consideration for our work and hope that any remaining concerns were alleviated and thus enabling you to support our paper more favorably.

---

### Official Review · Reviewer_VskD · 2021-11-02

**Correctness:** 3
**Technical Novelty And Significance:** 3
**Empirical Novelty And Significance:** 2
**Recommendation:** 6
**Confidence:** 3

**Main Review:**

Strengths:
- This paper proposed a new threat model called online adversarial attacks, which could open the door for a new research direction.
- Based on existing algorithms, the authors proposed Virtual+. This algorithm could also be used for tackling other online problems.
- The experiments show the proposed algorithm achieves better results than other baseline method.
- The theoretical analysis is thorough and technically sound.

Weaknesses:
- I feel hard to understand the motivation of studying online adversarial attack. I agree it is a new problem, but I am not sure about what kind of attacker need to launch online attack on what type of dataset? Please give a concrete example.
- The proposed method seems disconnected with the adversarial attack scenario. Which part of the Virtual+ algorithm is tailored for adversarial attacks? If the Virtual+ is a generalized algorithm which can be deployed on any k-secretary problem, why not position this paper as an algorithm paper instead of concentrating on adversarial attacks?
- The experiment only evaluated on simple image datasets like MNIST and CIFAR. This does not support the author's motivation that "Countless real-world applications involve streaming data that arrive in an online fashion (e.g., financial markets or real-time sensor networks)." Even for the image domain, MNIST and CIFAR-10 are considered to be insufficient.

**Summary Of The Paper:**

This paper proposed a new threat model for adversarial attacks called online adversarial attacks. Unlike traditional threat models, the online attack model assumes the data feed as stream and the decision made by the attacker are irrevocable. Towards this, the author made a connection between the online adversarial attack and the k-secretary problem, and proposed a new algorithm Virtual+. The experiments show that Virtual+ achieves better online fool rate than other baseline methods.

**Summary Of The Review:**

I am on the fence about this paper. On the one hand, the authors proposed a new threat model and provided with theoretically-grounded algorithm for tackling that problem. On the other hand, the motivation of the problem is not clear to me and the evaluation is also insufficient. I'd like to see other reviewer's opinion and the author's comment before making my final recommendation.

Post rebuttal:
I appreciate the author's effort on addressing my concerns and answering my questions. Most of my questions has been addressed. Thus, I'd like to raise my score to 6. However, I am still not fully convinced about the significance and motivation of studying online adversarial attack. My final recommendation is perhaps we could accept this paper and see how it can inspire future research on this direction.

---

> ### Author Response · Authors · 2021-11-17
> **Response to Reviewer VskD part 1/2**
>
> We would like to thank Reviewer VskD for their feedback. We appreciated that they think that our “new threat model called online adversarial attacks, [...] could open the door for a new research direction.” We now address their questions.
>
> Q: “but I am not sure about what kind of attacker need to launch online attack [..] Please give a concrete example.”
> A: We argue that in many real world situations (self driving cars, submission of images to a website, visual task for a robot) the data arrives to the model in an online fashion. Hence in order to minimize the odds to be detected, an attacker may want to have a budget on the number of attacks submitted. Our framework of online adversarial attack aims at formalizing this situation.
> A concrete example of such a task would be to attack the vision system of a robot or to attack the vision system of a surveillance camera.Please see the general comment to all reviewers for the details.
>
> Q: “The proposed method seems disconnected with the adversarial attack scenario”
> A: We appreciate the reviewer's concern but we politely disagree with this assertion that our algorithmic contributions are disconnected from the adversarial attack scenario. Specifically, we use secretary algorithms as a building block of our algorithmic contribution for Online Adversarial Attacks (Algorithm 2). Algorithm 2 is specific to online adversarial attack since it encompasses the fact that one does not have access to the target model to evaluate the potential of an attack. Thus, our method simultaneously tackles the challenges of online algorithms and transfer attacks. As we outline in our general comment to all reviewers online attacks can be viewed as a Man in the Middle attack adapted to deep learning systems which deepens the understanding of security vulnerabilities when working with streaming data. Specifically, we found that a few surprising phenomena (e.g. the relative strength of an attacker and the ease of attacking robust models) which are not realized in the whitebox nor as pronounced in the blackbox transfer setting. We believe the totality of this novel setting and the procedure to select highly susceptible data points for attack using secretary algorithms are tightly coupled with online adversarial attack threat models in the wild.
>
> Q: “ why not position this paper as an algorithm paper instead of concentrating on adversarial attacks?”
> A: We think that our contributions on the adversarial example side and on the online secretary algorithm side are both equally as important. The value we bring to the ICLR community is the *connection of two completely different communities*, namely the secretary algorithms and the adversarial examples communities. We believe *positioning this paper as an algorithm paper would undermine our main contribution* that is to define this new threat model and connect these two drastically different fields. We believe both aspects of our contributions have synergistic merits which is often realized when previous papers have successfully married two traditional different subfields towards an direction that is of interest to the ICLR and ML community at large. For example combining Group, Representation theory, and Deep Learning [6], Classical Graph Algorithms and Neural Nets [7], or Optimal Transport, Deep Learning and generative models [8].
>
> Response Part 1/2

---

> > ### Author Response · Authors · 2021-11-17
> > **Response to Reviewer VskD part 2/2**
> >
> > Q: “Even for the image domain, MNIST and CIFAR-10 are considered to be insufficient.”
> > A: For adversarial examples, one of the most challenging image datasets to attack is MNIST (e.g. the robust model by Madry et al. is very challenging to attack). It explains why MNIST and CIFAR, the two datasets on which challenges have been organized [1], are still meaningful baselines in adversarial robustness. We would like to point out that many papers published at the previous NeurIPS only considered MNIST and CIFAR (or SVHN, which is between MNIST and CIFAR in terms of complexity) datasets in their experiments [1,2,3,4].
> > Finally, we would like to point out that using standard, more straightforward vision datasets is common practice when introducing new settings. It was the case, for instance, in many lifelong continual learning papers [5]. Nevertheless we are currently in the process of running additional Imagenet experiments that we expect to report prior to the end of the discussion period.
> >
> >
> > [0] Madry, Aleksander, et al. "Towards Deep Learning Models Resistant to Adversarial Attacks." International Conference on Learning Representations. 2018.
> >
> > [1] Pal, Ambar, and Rene Vidal. "A Game Theoretic Analysis of Additive Adversarial Attacks and Defenses." Advances in Neural Information Processing Systems 33 (2020).
> >
> > [2] Jiang, Ziyu, et al. "Robust Pre-Training by Adversarial Contrastive Learning." NeurIPS. 2020.
> >
> > [3] Wu, Dongxian, Shu-Tao Xia, and Yisen Wang. "Adversarial Weight Perturbation Helps Robust Generalization." Advances in Neural Information Processing Systems 33 (2020).
> >
> > [4] Zhang, Richard Y. "On the tightness of semidefinite relaxations for certifying robustness to adversarial examples."NeurIPS 2020
> >
> > [5] Parisi, German I., et al. "Continual lifelong learning with neural networks: A review." Neural Networks 113 (2019): 54-71.
> >
> > [6] “Group Equivariant Convolutional Networks” by Cohen et. al ICML 2016.
> >
> > [7] “Neural execution of graph algorithms” by Velickovic et. al ICLR 2020
> >
> > [8] "Learning generative models with sinkhorn divergences." in AISTATS, Genevay et al. 2018.
> >
> > Response part 2/2

---

> ### Author Response · Authors · 2021-11-23
> **Kind Reminder to Respond to our Rebuttal**
>
> Dear Reviewer,
>
> We appreciate all of your comments and feedback provided in your review of our work. We wish to highlight that we have updated our manuscript and responded in detail to each point in our rebuttal below and addressed the motivation of online attacks in the general response to all reviewers. We believe we have successfully addressed the chief points of contention and we ask you for a renewed evaluation of our paper with the rebuttal, general response, and updated manuscript as context. We are also more than happy to answer any remaining questions you might have that could alleviate this process further.

---

> ### Author Response · Authors · 2021-11-29
> **Pleasant Reminder to respond to our rebuttal**
>
> Dear Reviewer,
>
> We thank you again for taking the time to read our responses to your questions. As you mentioned in your summary of your review that you are on the fence regarding this paper and that your overall evaluation will be based on specific answers to your questions and other reviews. We would like to point out that we have updated our manuscript with ImageNet results (Table 1) and as outlined in the general response to all reviewers the motivation for considering online attacks. Based on our rebuttal, general response and updated draft Reviewer 82UN upgraded their score from 6->8. We were wondering if our responses helped clear any remaining confusion, doubt or questions you might have so that you can potentially endorse our paper in a more favorable manner.

---

### Official Review · Reviewer_82UN · 2021-11-04

**Correctness:** 4
**Technical Novelty And Significance:** 3
**Empirical Novelty And Significance:** 3
**Recommendation:** 8
**Confidence:** 4

**Main Review:**

Strengths:
1. This paper is technically solid and detailed. It is in well structure and easy to follow. The whole story is complete with the theory supported.
2. The experiments are also good. Some findings in the paper do motivate us further reflecting on the difference of online attack and offline attack. This may provide a new perspective why deep learning model is so fragile.

However, i still have some concerns w.r.t. experiments.
1. As mentioned in the introduction, several works also study the online attack problem, more detailed description about the difference between their cases and this paper should be discussed, maybe in the related works? To be honest, i did not get the difference well.

2. Even previous works study a slightly different setting. Could their methods be adapted as more strong baselines for comparison?

3. How are the FGSM and PGD implemented in this paper？do we perform FGSM or PGD directly on the selected samples?

4. As discussed in Section 4, a surrogate function is used when the true model is unaccessible. How the surrogate function is selected in the experiment? have you tried different surrogate functions?

5. In the experiments,  counter-intuitive results are obtained. Have you tried different defence strategies for comparison?

6. I notice that the experiments are conducted on MNIST and CIFAR only, is the proposed method can be efficiently implemented on large-scaled dataset?

**Summary Of The Paper:**

This paper focuses on the online attack problem, where two elements are emphasized that attackers must operate under partial knowledge
of the target model, and the decisions made by the attacker are irrevocable. To solve this problem, they propose VIRTUAL+ and give theoretical guarantee on competitive ratio. The findings in the related experiments also validate the necessity and efficacy of proposed method in online setting.


**Summary Of The Review:**

This paper is solid and well written, however, there are some points i am confused. I thus give the score of 6.

====================================update========================================

First of all, I appreciate the authors' hard work in the rebuttal. Most of my concerns have been addressed in the discussion. I would like to raise the score to accept.

---

> ### Author Response · Authors · 2021-11-17
> **Response to Reviewer 82UN part 1/2**
>
> We thank the reviewer for their thoughtful comments and appreciate the fact the reviewer felt our main technical contributions were “solid, detailed, and the entirety of the story is complete”. We further appreciate that the reviewer felt that the paper was well structured with good experiments whose findings motivate the richness of our new proposed online threat model---leading to new perspectives on the fragility of deep learning models. We now address the main questions raised by the reviewer grouped by theme.
>
> **Comparison to prior work**
> We acknowledge the reviewer's comment regarding the lack of a clear distinction between prior work on similar attack settings, making it hard to place this work in the right context. We would like to first politely clarify---to the best of our knowledge---that our work is the only truly online threat model. We mean that our setting is the only one considering that data points are only ever observed once, and a decision to attack must be made at the moment and cannot be reversed retroactively. For example, Gong et. al 2019 [1] consider replay attacks on Voice-Controlled Systems whereby streamed audio input is captured with a recording device, and then the entire sequence is spoofed and replayed back. Unlike online attacks that we consider, they can manipulate the whole sequence retroactively and do not have to make an irreversible decision to attack at a given timestep. Similarly, both [2, 3] consider adversarial attacks against deep reinforcement learning agents. Like us, they consider an adversarial budget that limits the number of points to attack to avoid detection. However, unlike us, they require whitebox access to the target model in order to train another predictive model by interacting with the environment, which can later be used to inform “when to attack”. Thus the datapoints appearing at test time may already be seen and scored during the training period. The dichotomy between collecting data for potentially an infinite time horizon before attacking is at odds with our online threat model as a data point can only be observed once. As a result, none of these works can be used within our online threat model and are not appropriate baselines. We understand that this discussion was not sufficiently clear in the main draft and as a result we have updated the paper with an additional appendix G highlighting the technical difficulties of adapting prior work.
>
> **Implementation details**
> Q. “How are the FGSM and PGD implemented in this paper？do we perform FGSM or PGD directly on the selected samples?”
>
> The outline of how we perform online attacks is detailed in Algorithm 2 in the main paper. Concretely, given a fixed attack strategy (e.g. FGSM or PGD) the attack is conducted on all incoming data points but with respect to the surrogate model. Using the loss values incurred by the surrogate, an online algorithm decides whether the attack should then be submitted. Thus deciding when to attack in an online setting using secretary algorithms is the core technical contribution of this work.
>
> Q. “ How the surrogate function is selected in the experiment? have you tried different surrogate functions?”
>
> We thank the reviewer for raising this question. Please note that appendix E.2 outlines how exactly we chose our surrogate models and that we tried as much as possible to replicate the NoBox setup found in [4]. For example, in MNIST we consider an ensemble of 5 models (see Table 5 in the appendix for exact architecture details) from which the surrogate and target models are chosen without replacement. Similarly, for CIFAR10 both the surrogate and target models are selected from an ensemble of canonical architecture like VGG, WideResnet, and InceptionNet-V3. Please note that we considered five separate instantiations (e.g., random seed) for each architecture for diversity and error reporting.
>
> Response Part 1/2

---

> > ### Author Response · Authors · 2021-11-17
> > **Response to Reviewer part 2/2**
> >
> > **Counter-intuitive results**
> > We acknowledge the reviewer’s point that the counterintuitive results may be specific to adversarial training. It is an interesting point, but given the litany of defenses proposed in the literature, we believe adversarial training has withstood the test of time as a representative defense strategy on MNIST and CIFAR10 and were even the focus of the famous attack challenge [5]. We would like to highlight that Appendix F discusses one of the initially surprising results in detail in the difference between the online and offline settings for robust and non-robust models. The most salient points of the discussion are that if we consider the density ratio distribution of loss values of unsuccessful vs. successful attacks where we observe that significantly more data points incur high loss values but do not translate to successful attacks for non-robust models. Indeed, as online algorithms like Virtual+ operate purely based on surrogate loss values, attacking robust models is more challenging. Thus we find a more precise separation between high loss and successful attacks and low loss and unsuccessful attacks. We argue that this explanation accurately describes the phenomena we observe in our online setting.
> >
> > **Datasets**
> > While we consider experiments on both MNIST and CIFAR10 in this work we would like to alleviate the reviewer’s concern that our online algorithm faces no scalability issues to other larger datasets that are not already shared with adversarial attack strategies such as FGSM and PGD. As we also mention in our response to Reviewer Vskd, there are still many published papers at NeurIPS 2020 that only considered MNIST and CIFAR10 for their experiments [6,7,8,9]. However, to further alleviate the reviewers concern we are currently in the process of running additional ImageNet experiments which we expect to report prior to the end of discussion period.
> >
> > We would like to thank the reviewer for their review of our paper. We believe we have answered all the great points raised by the reviewer in our author response. We respectfully ask the reviewer to reconsider their impression of the paper and potentially improve the given score if the raised concerns have been allayed. We thank the reviewer again for their time and we are also happy to answer any further questions that arise.
> >
> >
> >
> > [1] Gong, Yuan, et al. "ReMASC: Realistic replay attack corpus for voice controlled systems." arXiv preprint arXiv:1904.03365 (2019)
> >
> > [2] Lin, Yen-Chen, et al. "Tactics of adversarial attack on deep reinforcement learning agents." arXiv preprint arXiv:1703.06748 (2017)
> >
> > [3] Sun, Jianwen, et al. "Stealthy and efficient adversarial attacks against deep reinforcement learning." Proceedings of the AAAI Conference on Artificial Intelligence. Vol. 34. No. 04. 2020
> >
> > [4] Bose, A. J., Gidel, G., Berard, H., Cianflone, A., Vincent, P., Lacoste-Julien, S., & Hamilton, W. L. (2020). Adversarial example games. NeurIPS 2020.
> >
> > [5] https://github.com/MadryLab/cifar10_challenge
> >
> > [6] Pal, Ambar, and Rene Vidal. "A Game Theoretic Analysis of Additive Adversarial Attacks and Defenses." Advances in Neural Information Processing Systems 33 (2020).
> >
> > [7] Jiang, Ziyu, et al. "Robust Pre-Training by Adversarial Contrastive Learning." NeurIPS. 2020.
> >
> > [8] Wu, Dongxian, Shu-Tao Xia, and Yisen Wang. "Adversarial Weight Perturbation Helps Robust Generalization." Advances in Neural Information Processing Systems 33 (2020).
> >
> > [9] Zhang, Richard Y. "On the tightness of semidefinite relaxations for certifying robustness to adversarial examples."NeurIPS 2020
> >
> > Response Part 2/2

---

> ### Author Response · Authors · 2021-11-23
> **Kind Reminder to Respond to our Rebuttal**
>
> Dear Reviewer,
>
> We are grateful for our feedback and comments that allowed us to strengthen our paper. We would like to highlight that we have updated our manuscript and responded to all of your key comments below in our rebuttal. We hope our response was satisfactory and aids you in potentially reevaluating our paper in a more favorable light. We would greatly appreciate a response from you and we are also happy to answer any remaining questions that you might have.

---

### Author Response · Authors · 2021-11-23
**Summary of our responses to the reviewers and main updates to paper**

We would like to thank all reviewers for their time and valuable feedback. We are grateful for their constructive criticisms that led us to improve the overall quality of our paper. We now summarize the main updates to the paper as well as key clarification points used in our rebuttal.

**Motivation for Online Attacks**

Please see our general comment to all reviewers where we highlight the importance of studying adversarial attacks in a truly online setting.

**Larger datasets for the experiments (*Reviewer 82UN*, *Reviewer VskD*):**

*Reviewer 82UN* and *Reviewer VskD* both pointed out the fact that our paper would benefit from some experiments on a larger dataset. We have added experiments on the *Imagenet* dataset for the experiments of Table 1. We find that our key findings on MNIST and CIFAR10 transfer over to Imagenet as well where we see that all online algorithms improve over the Naive baseline---note that the application of online algorithms for attack selection is also a key contribution of our work---and that Virtual+ is an especially strong secretary algorithm in terms of online fool rate.


 **Counterintuitive results (*Reviewer 82UN*, *Reviewer XV1X*):**
- *Reviewers 82UN and XV1X*: We have updated the caption in Fig 5 and the writing in Appendix F to further improve the description of the phenomena we observe for these counterintuitive results. We have also directly responded in detail to each of the reviewers' specific comments. We believe that our answers were satisfactory in exposing the nuance within the online setting. We are also happy to address any further questions the reviewers might have here.

- *Reviewer XV1X*: As suggested by the reviewer we have also tried to re-run PGD attack with n_iter = 100 on Imagenet. As we can see, these experiments confirm the point we make in our rebuttal to the review: “iterated attacks lead to less transferable attacks [5,6,7]” and this phenomena is further exacerbated in the online setting (please find the numbers directly in our rebuttal).


**Description of Prior Work Reviewer 82UN)**

We appreciated the reviewers comment on adding more description on prior work in related settings. To this end, we have added a new Appendix G for the key prior works and how they are different from our truly online adversarial attack setting.

We hope that these explanations and resulting updates to the paper alleviates the main points raised by the reviewers and also allow them to endorse the paper more wholeheartedly.


[1] Gong, Yuan, et al. "ReMASC: Realistic replay attack corpus for voice controlled systems." arXiv preprint arXiv:1904.03365 (2019)

[2] Lin, Yen-Chen, et al. "Tactics of adversarial attack on deep reinforcement learning agents." arXiv preprint arXiv:1703.06748 (2017)

[3] Sun, Jianwen, et al. "Stealthy and efficient adversarial attacks against deep reinforcement learning." Proceedings of the AAAI Conference on Artificial Intelligence. Vol. 34. No. 04. 2020

---

### Decision · Program_Chairs · 2022-01-20

**Decision:**

Accept (Poster)

**Comment:**

This paper opens the area of adversarial-attack research on streaming data (e.g., real-world settings such as self-driving cars and robotic visual tasks for a robot). For instance, online adversaries can focus their attack on a small subset of the streamed/online data, but still cause much damage to downstream models. This work highlights the need for stateful defense strategies. Connections to online algorithms and the k-secretary problem are made, along with improvements to some online-algorithms work of Albers and Ladewig.

Overall, the attack model introduced is important, and the bridge to online algorithms would be useful for the ICLR community. I also believe this topic lends diversity to the typical set of ICLR papers.